# Detection of Amyotrophic Lateral Sclerosis (ALS) Comorbidity Trajectories Based on Principal Tree Model Analytics

**DOI:** 10.3390/biomedicines11102629

**Published:** 2023-09-25

**Authors:** Yang-Sheng Wu, David Taniar, Kiki Adhinugraha, Li-Kai Tsai, Tun-Wen Pai

**Affiliations:** 1Department of Computer Science and Information Engineering, National Taipei University of Technology, Taipei 106, Taiwan; t110598007@ntut.org.tw; 2Department of Software Systems & Cybersecurity, Monash University, Melbourne, VIC 3800, Australia; david.taniar@monash.edu; 3Department of Computer Science and Information Technology, La Trobe University, Melbourne, VIC 3086, Australia; k.adhinugraha@latrobe.edu.au; 4Department of Neurology and Stroke Center, National Taiwan University Hospital and National Taiwan University College of Medicine, Taipei 100, Taiwan; milikai@ntuh.gov.tw

**Keywords:** Amyotrophic Lateral Sclerosis (ALS), principal tree, disease progression, clinical data, disease trajectory

## Abstract

The multifaceted nature and swift progression of Amyotrophic Lateral Sclerosis (ALS) pose considerable challenges to our understanding of its evolution and interplay with comorbid conditions. This study seeks to elucidate the temporal dynamics of ALS progression and its interaction with associated diseases. We employed a principal tree-based model to decipher patterns within clinical data derived from a population-based database in Taiwan. The disease progression was portrayed as branched trajectories, each path representing a series of distinct stages. Each stage embodied the cumulative occurrence of co-existing diseases, depicted as nodes on the tree, with edges symbolizing potential transitions between these linked nodes. Our model identified eight distinct ALS patient trajectories, unveiling unique patterns of disease associations at various stages of progression. These patterns may suggest underlying disease mechanisms or risk factors. This research re-conceptualizes ALS progression as a migration through diverse stages, instead of the perspective of a sequence of isolated events. This new approach illuminates patterns of disease association across different progression phases. The insights obtained from this study hold the potential to inform doctors regarding the development of personalized treatment strategies, ultimately enhancing patient prognosis and quality of life.

## 1. Introduction

Amyotrophic Lateral Sclerosis (ALS) is a rare but devastating neurodegenerative disease that affects upper and lower motor neurons, primarily observed in men. The disease is typically diagnosed around the age of 62, and the median survival time for patients with ALS is approximately three years [1]. Patients often succumb to respiratory complications such as bronchopneumonia or pneumonia [2,3,4] In Taiwan, the average annual incidence and prevalence rates were 0.51 and 1.97 per 100,000 individuals, respectively. Furthermore, the economic burden of ALS management was significantly higher than average healthcare costs [5]. Despite the diverse clinical phenotypes of the disease, the pathophysiology of ALS remains unknown [2]. Although substantial efforts have been made, viable interventions to halt or slow down the progression of the disease remain elusive, underscoring the inherent complexity of ALS [6].

Studies have found comorbidities in patients with ALS at frequencies that differ from the general population [7]. These comorbidities may modulate disease progression. For example, diabetes and hypertension correlated with delayed onset of ALS, while specific cardiovascular disorders could influence the course of the disease positively or negatively [7,8,9,10]. However, most of these potential risk factors related to ALS were often assessed in isolation, stressing the need for an integrated approach to understand the intricate relationships between comorbidities and ALS [11].

Given the progressive nature of ALS, understanding the comprehensive course of the disease is crucial for optimal clinical management [12]. However, many studies often restricted the prediction of the disease to a single progression event or categorize patients into static progression phenotypes [13,14]. Such methodologies fall short of capturing the dynamic narrative of disease progression and interactions with comorbidities. Furthermore, traditional analysis methods [15,16,17,18,19,20] often interpreted disease trajectories as a chain of disease events, potentially overlooking the simultaneous presence of comorbidities in ALS patients.

Recent developments in unsupervised machine learning algorithms such as Elastic Principal Graphs (ElPiGraph) have paved the way for a more comprehensive analysis of complex multidimensional data. These techniques, widely applicable in various fields including biology and medicine, offer opportunities to construct robust and scalable models of intrinsic dataset geometry. This enables the identification and analysis of high-dimensional data points and their respective relationships [21].

Our study leverages the capabilities of the ElPiGraph method to unravel the associations between ALS and its comorbidities. We introduce a novel perspective that views ALS trajectories as a succession of disease stages characterized by cumulative disease occurrences and transitions. By transforming patient records into the same multidimensional space, our approach aims to bridge the gap between individual patient experiences. It provides a more comprehensive and nuanced interpretation of ALS progression and extracts more insightful knowledge from the available data.

## 2. Materials and Methods

### 2.1. Taiwan’s ALS Dataset

This study utilized anonymized data from 83 patients diagnosed with Amyotrophic Lateral Sclerosis (ALS) obtained from a cohort dataset of one million enrollees of Taiwan’s National Health Insurance Research Database (NHIRD) over an 18-year span (1996–2013) [22]. Of these patients, 51 (61.45%) were male and 32 (38.55%) were female. The age of their first ALS diagnosis ranged from 34 to 70 years, with an interquartile range (IQR) between 44 and 62 years. The dataset includes 50,613 related medical records documenting hospital, outpatient, and emergency visits for the 83 ALS patients. Each record includes the date and diagnosis, as represented by the International Classification of Diseases, Ninth Revision (ICD-9) [23].

Given the heterogeneity of ALS, we focused on cases representing definitive diagnoses involving both upper and lower motor neurons (ICD-9 335.20). Alongside these cases, we assembled a control group consisting of four-fold subjects with age- and gender-matched individuals for each ALS patient within a three-year age range (1000 days). This control group consisted of 332 patients without ALS, with a total of 158,677 corresponding medical records.

### 2.2. Study Design

This study consisted of four key stages for trajectory analysis based on principle tree approaches, including the following:transformation of patient medical records into a quantifiable multi-dimensional space;application of principal trees to reveal the concealed data geometry and topology;identification of diseases associated with ALS across different pathways and stages to highlight comorbidity interaction;extraction of different trajectories from the ALS patient group progressing to various final states.

#### 2.2.1. Transforming Patient Medical Records into a Multidimensional Space

Each medical record includes three essential elements: (1) patient identifier (PID), (2) visit date, and (3) diagnosis of the disease according to ICD-9. We transformed each diagnosis into a one-hot encoding format, which is a binary representation where each unique diagnosis was encoded as a separate feature. The evolution of diseases was then reflected based on each patient’s cumulative count of disease occurrences (we counted each diagnosis once per patient per day).

The process of transforming patient medical records into a multidimensional space is outlined in Figure 1. The original table of patient records was first converted into a one-hot table, where each unique diagnosis was represented by its own column. For example, the number “1” indicates the presence of the diagnosis for a given patient on a particular date, while a “0” indicates its absence. This one-hot table was then transformed into its corresponding cumulated version, accumulating the count of each diagnosis over time for each patient.

We standardized the features by scaling and centering the data to unit variance. We then conducted a principal component analysis (PCA) using 60 components, as determined by the elbow rule—a method used to choose the optimal number of components by locating the “elbow” in the plot of explained variance versus the number of components [24,25]. This analysis transformed 50,613 initial records for 83 ALS patients with 493 unique diagnoses into 22,981 records in 60 dimensions.

#### 2.2.2. Application of Principal Tree to Reveal ALS Progression Pathway

A principal tree serves as an effective data approximator, capturing complex relationships within a data space that may not be attainable with linear segments alone [26]. It helps reveal the underlying structure in the disease progression data that might remain obscured and ignored. We utilized the ElPiGraph library [21] to construct the principal trees, aiming to grasp the intricate topology and progression patterns of ALS.

Once the principal tree was constructed, we analyzed the directionality of the edges connecting the nodes. This analysis enables us to map the trajectories representing the progression of ALS. We designated the root node as the node representing the state before any diagnosis, where all cumulative counts were initially zero. As we moved further away from the root node, the nodes represented progressively advanced stages of the disease for ALS subjects, and we could determine the directions of edges accordingly. The terminal states of the trajectories were then marked as the tree’s leaf nodes.

#### 2.2.3. Exploring Disease Associations and Comorbidity Interaction in ALS

To differentiate between disease progression and various pathways, we adopted an approach that located diseases during state transitions. This strategy involved three main tasks: detection of potential diseases that either contribute to or arise from ALS, extraction of diseases that show increased occurrences during state transitions, and identification of unique diseases associated exclusively with each branch.

To identify diseases significantly associated with ALS, we used the odds ratio, a statistical measure widely employed in epidemiological studies. The odds ratio assesses the likelihood of an event occurring in one group compared to another, thereby indicating the effect size [27,28]. In our analysis, we set an odds ratio threshold of 2 and required a 95% confidence interval that did not include zero to ensure the clinical significance of the identified associations [29,30].

Subsequently, we identified diseases associated with each edge of the principal tree, specifically those where the number of diagnoses increased during a state transition. This approach does not only highlight disease dynamics but also helps characterize different progression pathways. Additionally, we pinpointed “branch-only” diseases, those unique to a particular branch and not inherited from ancestral branches or shared with others. Identifying these unique diseases enriches our understanding of distinct disease interactions within each trajectory.

#### 2.2.4. Extraction and Classification of ALS Disease Trajectories

We constructed disease trajectories from previous principal branches, which are paths that extend from the root node to the leaf nodes in the multidimensional space defined by cumulative counts of disease occurrences. Each patient’s final medical record is transformed into a point within this space. The “nearest” node, determined by the shortest geometric distance, is used to map each patient’s final observed state within this network of disease trajectories. If a patient’s last observed state aligns with a single trajectory, we assigned them to that specific category. However, if the patient’s final state overlapped with multiple trajectories—potentially due to truncated observable trajectories—we assigned these patients to an uncategorized group for further analysis.

## 3. Results

### 3.1. Identification of Diseases Associated with ALS

Our analysis identified several diseases with a significant association with Amyotrophic Lateral Sclerosis (ALS), as indicated by odds ratios (ORs) and 95% confidence intervals (CI) displayed in Table 1. Each listed disease varies in ORs, highlighting the relative likelihood of simultaneous diagnosis in ALS patients compared to non-ALS individuals.

Of these diseases, 28 were present at one or more state transition edges, suggesting their potential involvement in disease progression. Additionally, 26 diseases were designated as “branch-only”, appearing exclusively within certain branches and not being traceable to ancestral lines. These “branch-only” diseases suggest distinctive ALS-related disease pathways. Furthermore, 23 diseases were linked exclusively to a specific edge, reinforcing the idea of specialized disease trajectories.

Seven diseases showed an infinite OR, implying a solid association with ALS. These included “dislocation of the jaw”, “chromosomal anomalies”, “late effects of acute poliomyelitis”, “acute poliomyelitis”, “encephalitis myelitis and encephalomyelitis”, “muscular dystrophies and other myopathies”, and “disorders of other cranial nerves”.

Other diseases, such as “other diseases of spinal cord” and “Parkinson’s disease”, demonstrated high ORs, indicative of a significant connection with ALS. “Malignant neoplasm of prostate” and “diseases of white blood cells” also suggested a meaningful relationship with ALS, despite broad 95% CIs.

Several neurological conditions, including “hereditary and idiopathic peripheral neuropathy”, “infantile cerebral palsy”, and “other conditions of brain”, showed strong associations with ALS, reinforcing its neurological nature. Interestingly, conditions not directly related to the nervous system, like “septicemia”, “infections of kidney”, and “cardiac dysrhythmias”, were also associated with ALS, pointing to the disease’s potential systemic impacts.

### 3.2. Decoding ALS Pathways through the Principal Tree

When exploring the pathways of progression of ALS, we used a principal tree to capture the courses of disease progression. As shown in Figure 2, the tree begins with a root node (0), representing the state of the disease prior to diagnosis. The tree then bifurcates into multiple pathways, ending in 12 leaf nodes, each representing a distinct potential path of disease progression.

All patients in our study started at root node 0 and advanced to the first bifurcation at node 1, marking the onset of ALS. This transition was characterized by an increased prevalence of nine diseases, indicating that these may be early indicators of the progression of ALS. Beyond node 1, the tree split into four distinct pathways, each leading to different outcomes in the progression of the disease. The distribution of diseases and patients across unique pathways in the ALS progression principal tree were shown in Table 2.

The paths from node 1 to node 9 and from node 1 to node 2 did not show “branch-only” diseases, suggesting a common progression route for many ALS patients without developing specific diseases. However, the routes from node 1 to nodes 4 and 3 displayed unique “branch-only” diseases, hinting at alternative pathogenetic pathways. These included “Dementia” (290) on the path to node 4, while “Acute poliomyelitis” (045), “Muscular dystrophies and other myopathies” (359), and “Other paralytic syndromes” (344) were on the path to node 3.

The branch from node 3 further bifurcates into multiple pathways, each marked by different “branch-only” diseases. For example, the path from node 3 to node 8 was distinguished by “infantile cerebral palsy” (343), while the route from node 3 to node 5 was identified by “Facial nerve disorders” (351). Furthermore, the transition from node 3 to node 6 was marked by “Other unspecified disorders of the nervous system” (349), “Mononeuritis of the lower extremity and an unspecified site” (355), and “Encephalitis myelitis and encephalomyelitis” (323).

Collectively, this principal tree model offers a visualization of the varied trajectories of progression of ALS, suggesting that, while some patients follow common progression routes with no unique “branch-only” diseases, others may experience specialized pathways with distinct diseases marking their journeys.

### 3.3. Delineation and Analysis of Patient Trajectories in ALS Progression

Our study yielded the delineation of eight distinct trajectories representing ALS progression patterns, each substantiated by at least three patients. The varied patient trajectories in the ALS progression principal tree were shown in Figure 3 respectively. These trajectories revealed the dynamic landscape of ALS progression.

We identified a distinctive “Uncategorized Group” comprising patients whose progression patterns do not fit within any of the eight defined trajectories. These patients’ disease pathways may be shorter, failing to exceed a bifurcation point that leads to only a terminal node, suggestive of a potentially truncated ALS progression pattern. This unique progression pattern, as observed in the initial ALS records for these patients, tends to cluster at the early stages of the disease. This divergence warrants further investigation and may provide valuable insights into the heterogeneity of ALS progression.

Apart from the Uncategorized Group, our analysis classified patients into seven distinct trajectories, each reflecting the unique progression dynamics of ALS. The diversity in these trajectories echoes the high heterogeneity in the progression of ALS, underscoring the potential of distinct underlying disease mechanisms or risk factors.

## 4. Discussion

Amyotrophic Lateral Sclerosis (ALS) is a rapidly progressing neurodegenerative condition characterized by complex and multifaceted dynamics, which pose challenges to our understanding of its progression. In order to address the complexities, our study employed a data-driven approach to unravel the intricate patterns of ALS progression and its interplay with concurrent diseases.

We utilized a principal tree algorithm to capture the underlying data geometry forming the multifaceted ALS progression patterns. In our framework, ALS progression is conceptualized as a journey through sequential stages of the disease, each represented by a node in the tree. Each node signifies an accumulation of diseases, while the edges connecting nodes symbolize potential transitions, thus depicting the dynamic nature of disease progression and the directional shifts that reflect ALS progression in relation to cumulative comorbidities.

While the traditional literature often portrays disease trajectories as sequences of discrete disease events, our model offers a fresh perspective by viewing trajectories as progressions through stages, each comprising multiple co-existing conditions. This holistic view fosters a more genuine and subtle understanding of ALS progression, revealing the intricate relationship between ALS and its comorbid conditions.

One significant aspect of our study is the exploration of ALS’s relationship with concurrent diseases. Conventional research methods typically analyze ALS-associated risk factors independently, frequently overlooking the complex interactions among comorbid conditions. In contrast, our study contextualizes ALS within a network of co-existing diseases, underscoring the potential influence of these conditions on ALS progression. Our methodology, which converts patient records into a multidimensional space, enables a deeper understanding of disease progression by visualizing all disease states as interconnected components of a complex system.

For instance, from node 0 through node 1 to node 3, our analysis elucidates the common pathway of early symptoms encountered during the initial stages of ALS.

The early diagnosis of ALS presents a challenge due to the disease’s heterogeneity, subtle onset, and overall rarity. This often results in misdiagnoses that can prompt unsuitable treatments, such as unneeded surgeries, leading to delays in proper diagnosis and exacerbating patient morbidity [31,32,33].

In the transition from node 0 to node 1, the majority of misdiagnoses are “disorders of the peripheral nervous system (350–359)”, “dorsopathies (720–724)”, and “disorder of soft tissue (725–729)”. These non-specific conditions, often manifesting with symptoms such as muscle weakness and fatigue akin to ALS, highlight musculoskeletal and peripheral nervous system disorders in early ALS misdiagnosis.

Interestingly, the prevalence of misdiagnoses such as “anxiety, dissociative and somatoform disorders (300)” suggests a tendency towards psychosomatic interpretations of early ALS symptoms.

The shift from node 1 to node 3 sees a transition towards diseases with more defined neurological characteristics, such as pure motor weakness of limbs, muscle atrophy, and fasciculations. These conditions often share overlapping neurological symptoms with ALS, such as muscle atrophy, fasciculations, and limb weakness, contributing to misdiagnoses.

Our analysis captures the complexities of diagnosing ALS, considering the myriad conditions that can mimic its early symptoms. It emphasizes the necessity for a more sophisticated and holistic diagnostic approach that encompasses a robust understanding of ALS’s varied presentations and promotes interdisciplinary collaboration among general practitioners, neurologists, and other specialists.

Building on the foundation of our computational findings, it becomes imperative to consider integrating complementary experimental techniques, for example, in the domain of soft matter nanomechanics. Techniques such as atomic force microscopy (AFM) and optical tweezers, highlighted by Magazzù et al. [34], stand out as invaluable tools in shedding light on the biomechanical intricacies at the single-molecule level. These methods, by quantifying nanomechanical parameters like Young’s modulus, hold the potential to enrich our understanding of cellular systems, especially within the scope of neurodegenerative pathologies. Likewise, Strijkova et al. [35] have intelligently leveraged AFM to unravel the distinctive mechanical and morphological attributes of platelets in various neurodegenerative conditions, including Parkinson’s disease, ALS, and Alzheimer’s disease. Such groundbreaking experimental works underscore the imperative of meshing computational analyses with tangible experimental data. Furthermore, Varga et al. [36] employed AFM to delve into the elasticity of ALS myotubes, attributing it to a specific ALS-inducing mutation in mice. These revelations can potentially layer our understanding of ALS, bridging its pathogenesis to cellular manifestations.

While our study offers novel insights into the complexities of ALS progression, it is crucial to validate the robustness and reliability of our findings further. Future research must verify the consistency of our model’s representation of disease trajectories across various datasets and diverse patient demographics. Despite these considerations, our study introduces a pioneering approach of incorporating comorbidity data into tree edges during disease stage transitions, offering a fresh perspective on understanding ALS. We suggest that this approach has the potential to considerably enhance our comprehension of ALS and provide a solid analytical framework for similar future research. Through these contributions, we aspire to promote the development of improved care strategies and therapeutic interventions for this debilitating disease.

## Figures and Tables

**Figure 1 biomedicines-11-02629-f001:**
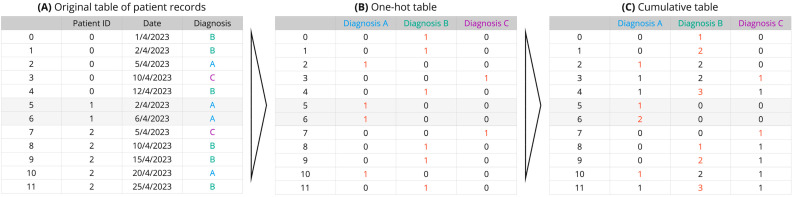
Schematic representation of transforming patient medical records into a multidimensional space. The process consists of three steps. (**A**) The original table of patient records, with each record featuring the patient identifier (PID), the hospital visiting date, and the disease diagnosis. (**B**) Conversion of the original table into a one-hot table, with each unique diagnosis represented as a separate column. In this format, a ‘1′ signifies the presence of the diagnosis for a given patient on a particular date, while a ‘0′ denotes its absence. (**C**) Transformation of the one-hot table into a cumulative table, aggregating the occurrence count of each diagnosis over time for each patient. This cumulative representation forms the basis for further multidimensional data analysis.

**Figure 2 biomedicines-11-02629-f002:**
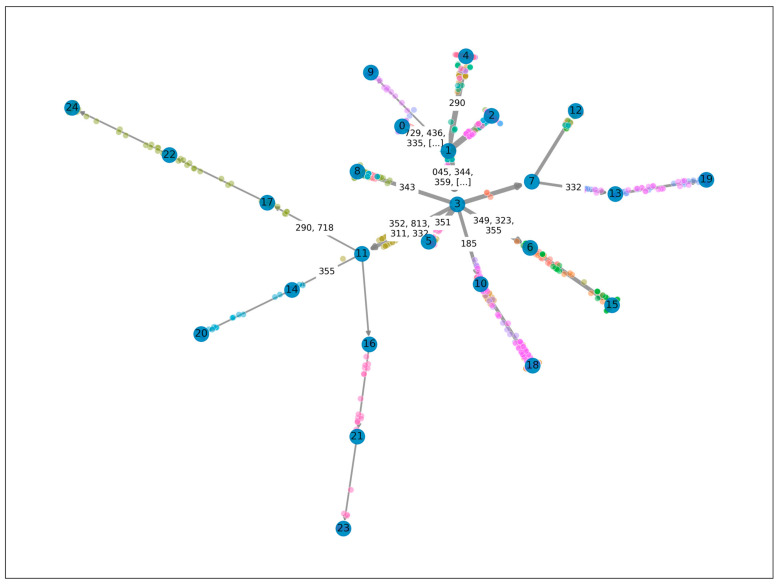
Principal tree illustrating heterogeneous pathways of progression of ALS disease. The tree originates from a root node labeled as “0”, and it is split into multiple pathways that end at twelve leaf nodes. Each leaf node represents a unique trajectory of progression of ALS. The tree captures the presence of “branch-only” diseases along specific routes, highlighting distinct disease progression pathways. The numbers shown within the nodes serve as identifiers and possess no intrinsic meaning. The nodes are linked by edges, the width of edge corresponding to the number of patients following a particular pathway. The presence of a disease is indicated by its ICD-9 code. Points scattered throughout the figure represent individual patient medical visits, color-coded for patient differentiation.

**Figure 3 biomedicines-11-02629-f003:**
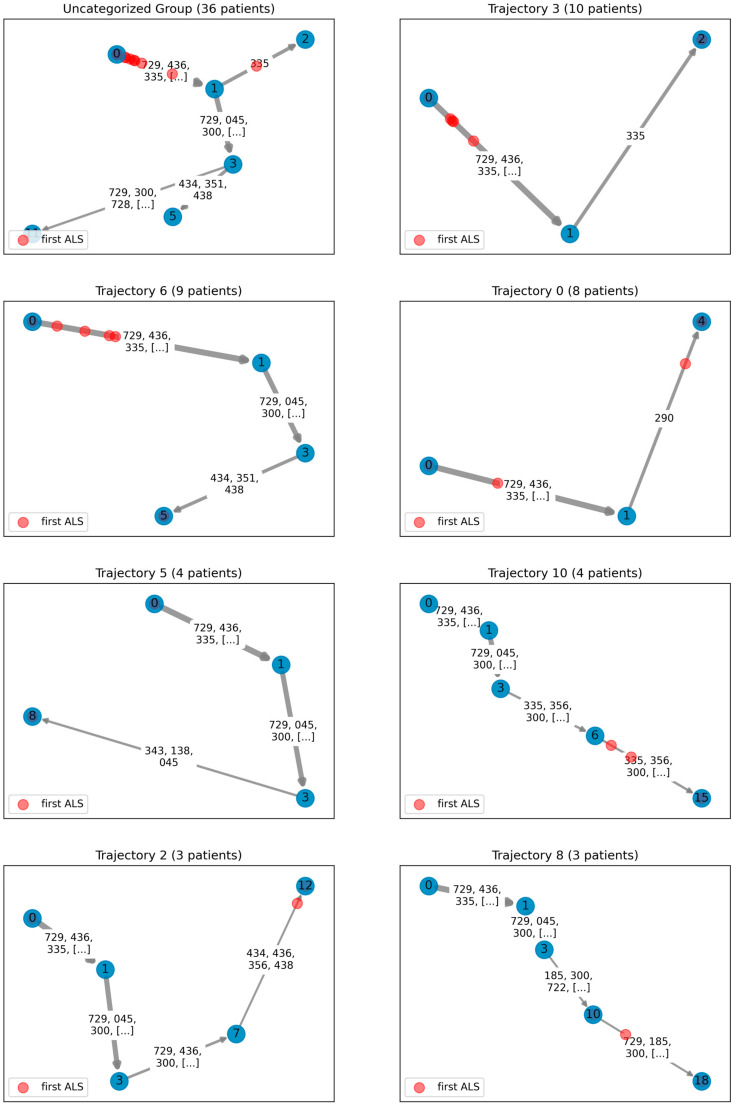
Detailed illustration of varied patient trajectories in the ALS progression principal tree. The figure consists of eight subgraphs, each representing a unique disease progression pathway undertaken by a specific patient group, originating from the initial onset (node 0) to differing terminal stages. Subgraphs are sorted based on the number of patients in each group. The “Uncategorized Group” encompasses patients whose progression patterns do not fit into any trajectory. The trajectories of these patients may be too short, failing to surpass a bifurcation point leading to a terminal node, indicative of a potentially truncated ALS progression pattern. Each of the subsequent subgraphs elucidates distinct ALS progression dynamics, characterized by changes in associated diseases (denoted by ICD-9 codes) along each pathway. The red points across the pathways denote the location of the first ALS onset for each patient, highlighting the varied progression patterns within ALS. The title of each subgraph specifies the particular trajectory along with the respective patient count.

**Table 1 biomedicines-11-02629-t001:** Diseases significantly associated with Amyotrophic Lateral Sclerosis (ALS). Diseases are denoted by the International Classification of Diseases (ICD-9) codes and listed alongside their respective odds ratios (ORs) and 95% confidence intervals (CIs).

ICD-9	Disease Name	OR	95% CI
038	Septicemia	2.85	1.28–6.17
045	Acute poliomyelitis	inf	1.68–inf
138	Late effects of acute poliomyelitis	inf	6.10–inf
185	Malignant neoplasm of prostate	16.6	1.61–827.73
242	Thyrotoxicosis with or without goiter	3.35	1.26–8.64
276	Disorders of fluid electrolyte and acid–base balance	4	1.86–8.54
288	Diseases of white blood cells	16.6	1.61–827.73
290	Dementias	4.39	1.58–12.24
300	Anxiety, dissociative, and somatoform disorders	2	1.20–3.37
311	Depressive disorder, not elsewhere classified	3.23	1.16–8.71
323	Encephalitis myelitis and encephalomyelitis	inf	1.68–inf
331	Other cerebral degenerations	5.07	1.25–21.58
332	Parkinson’s disease	18.35	4.78–103.91
333	Other extrapyramidal disease and abnormal movement disorders	4.44	1.67–11.78
336	Other diseases of spinal cord	24.93	5.28–236.30
343	Infantile cerebral palsy	8.48	1.77–53.57
344	Other paralytic syndromes	5.1	2.34–11.17
345	Epilepsy and recurrent seizures	3.29	1.01–10.30
348	Other conditions of brain	7.9	2.30–30.90
349	Other and unspecified disorders of the nervous system	3.66	1.85–7.18
351	Facial nerve disorders	6.98	1.33–45.92
352	Disorders of other cranial nerves	inf	2.70–inf
353	Nerve root and plexus disorders	5.4	2.97–9.83
354	Mononeuritis of upper limb and mononeuritis multiplex	3.33	1.76–6.23
355	Mononeuritis of lower limb and unspecified site	3.82	1.54–9.34
356	Hereditary and idiopathic peripheral neuropathy	15.36	6.77–37.64
357	Inflammatory and toxic neuropathy	6.82	2.92–16.47
358	Myoneural disorders	7.86	3.08–21.27
359	Muscular dystrophies and other myopathies	inf	16.82–inf
360	Disorders of the globe	4.93	1.51–16.50
427	Cardiac dysrhythmias	2.11	1.17–3.73
434	Occlusion of cerebral arteries	3.15	1.64–5.99
436	Acute, but ill-defined, cerebrovascular disease	5.06	2.28–11.32
438	Late effects of cerebrovascular disease	2.59	1.21–5.39
459	Other disorders of circulatory system	4.98	1.39–18.48
518	Other diseases of lung	6.28	2.73–14.81
590	Infections of kidney	2.75	1.01–7.14
714	Rheumatoid arthritis and other inflammatory polyarthropathies	2.49	1.10–5.42
718	Other derangement of joint	3.22	1.28–7.86
721	Spondylosis and allied disorders	2.35	1.40–3.99
722	Intervertebral disc disorders	2.18	1.28–3.71
728	Disorders of muscle ligament and fascia	2.49	1.46–4.23
729	Other disorders of soft tissues	3.59	1.70–8.47
758	Chromosomal anomalies	inf	1.68–inf
813	Fracture of radius and ulna	3.42	1.13–10.00
830	Dislocation of jaw	inf	1.68–inf
891	Open wound of knee, leg (except thigh), and ankle	2.18	1.08–4.29
952	Spinal cord injury without evidence of spinal bone injury	8.48	1.77–53.57

**Table 2 biomedicines-11-02629-t002:** Distribution of diseases and patients across unique pathways in the ALS progression principal tree. Each row represents an edge (transition between nodes) within the ALS progression tree model. Diseases are marked by ICD-9 codes and categorized into “Associated Diseases” and “Branch-Only Diseases”. “Associated Diseases” represent all illnesses observed along a specific edge, whereas “Branch-Only Diseases” specify those unique to a specific edge, exclusive of those shared with other branches originating from the same parent node or inherited from ancestral nodes. The “Number of Patients” column quantifies the patients traversing each pathway, presented as both a numerical count and a percentage relative to the total patient cohort. This table underscores the complex intertwining of disease states across varying ALS progression paths and emphasizes the disease’s heterogeneity.

Edge	Associated Diseases (ICD-9)	Branch-Only Diseases (ICD-9)	Number of Patients
1–3	045, 729, 722, 721, 359, 353, 438, 427, 344, 300, 728, 138, 357	045, 359, 438, 427, 344, 138	31 (37.35%)
1–4	290	290	8 (9.64%)
1–9	721, 729, 722, 436		1 (1.20%)
1–2	335		10 (12.05%)
3–11	332, 729, 722, 721, 352, 813, 353, 344, 300, 728, 311	352, 813, 332, 311	3 (3.61%)
3–7	729, 356, 438, 427, 300, 434, 436		5 (6.02%)
3–8	343, 045, 138	343	6 (7.23%)
3–5	434, 351, 438	351	9 (10.84%)
3–10	722, 359, 353, 427, 300, 185	185	3 (3.61%)
3–6	335, 721, 356, 355, 300, 323, 349, 357	349, 355, 323	4 (4.82%)
11–17	332, 718, 300, 290, 813	290, 718	1 (1.20%)
11–14	729, 722, 721, 353, 344, 355, 300, 728, 311	355	1 (1.20%)
11–16	352		1 (1.20%)
7–12	434, 356, 436, 438		3 (3.61%)
7–13	335, 332, 729, 359, 438, 427, 300	332	2 (2.41%)
0–1	335, 729, 722, 721, 353, 300, 728, 357, 436	335, 729, 722, 721, 353, 300, 728, 357, 436	55 (66.27%)
17–22	332, 718, 300, 290, 813		1 (1.20%)
22–24	332, 718, 300, 290, 813		1 (1.20%)
14–20	729, 722, 721, 353, 344, 355, 300, 728, 311		1 (1.20%)
16–21	352		1 (1.20%)
10–18	729, 722, 359, 353, 427, 300, 185		3 (3.61%)
21–23	352		1 (1.20%)
6–15	335, 721, 356, 355, 300, 323, 349, 357		4 (4.82%)
13–19	335, 332, 729, 359, 438, 427, 300, 434		2 (2.41%)

## Data Availability

The datasets are from the NHIRD and were used only for this study under policy limitation. Therefore, the data are not publicly available. Data usage regulations can be checked from the website https://nhird.nhri.org.tw/en/ (accessed on 1 September 2023).

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
