# Peer review of "Detection of Amyotrophic Lateral Sclerosis (ALS) Comorbidity Trajectories Based on Principal Tree Model Analytics"

_biomedicines, 2023, doi:10.3390/biomedicines11102629_

Round 1

Reviewer 1 Report

The manuscript by Wu et al. is focused on the mapping ALS trajectories and clinical diagnoses using a principal tree-based model. The manuscript contains some novel data, is well written and easy to follow. Nonetheless, I would like to encourage the authors to think about the title. The current one is very broad. In my opinion it should be more specific demonstrating the major result(s) of the study.

Author Response

Response: Thanks reviewer’s comments. We appreciate for the suggestion of naming a new title for this paper. We have revised the paper title to reflect the major outcomes of this study in a better way. Here is the revised title: “Detection of Amyotrophic Lateral Sclerosis (ALS) Comorbidity Trajectories based on Principal Tree Model Analytics”.

Reviewer 2 Report

The manuscript titled “Mapping ALS Trajectories and Clinical Diagnoses: Insights from a Principal Tree-Based Model” by Wu, Y-S.; et al. is an original work where the authors assessed the impact of comorbilities on amyotrophic lateral diseases (ALS). The algorithm used by the authors served to decipher the how the progress of ALS is affected by these comorbilities. For it, the data from 83 individual were used. This number provides accurate statistical analysis and the subsequent accuracy of the most relevant findings. The scientific content is interesting and the sections are well-designed.

However, it exists some points that need to be addressed (please, see them below detailed point-by-point). The most relevant outcomes found by the authors can contribute to in the growth of many fields like the clinical&healthcare overall related to prognose not only ALS but also many other neurodegenerative disorders. For this reason, I will recommend the present scientific manuscript for further publication in Biomedicines once all the below described suggestions will be properly fixed.

Here, there exists some points that must be covered in order to improve the scientific quality of the manuscript paper:

1) TITLE. (OPTIONAL) The authors should consider add the full name “amyotrophic lateral sclerosis” instead of its abbreviated form.

2) INTRODUCTION. “Amyotrophic Lateral Sclerosis (ALS) (…) is approximately three years” (lines 30-33). Here, it lacks a relevant reference related to the information content provided by the authors [1].

[1] Mead, R.J.; Shan, N.; Reiser, H.J.; Marshall, F.; Shaw, P.J. Amyotrophic lateral sclerosis: a neurodegenerative disorder poised for successful therapeutic translation. Nat. Rev. Drug Discov. 2023, 22, 185-212. https://doi.org/10.1038/s41573-022-00612-2.

3) MATERIALS & METHODS. “This study anonymized data from 83 patients (…) over an 18-years span (1996-2013)”. Please, the authors should provide further statistic insights about  the sex and age of the individuals who have participated in this study.

4) Figure 1 (line 109). The lettering related to this Figure should be enlarged. This will significantly aid potential readers to better visualized the information furnished by the authors. Same comment for Figure 2 (line 228) and Fig. 3 (line263).

5) Table 1 (line 193). Why did not the authors list the disease numbers by ICD-9 numerical order?

6) DISCUSSION. The authors perfectly highlights the most relevant outcomes found in this work. Nevertheless, since this study is mainly focused on computational studies the authors need to add some experimental approaches which could be complementary with this current research. In this context, the use of single-molecule techniques [2] is relevant to unravel the nanomechanical fingeprint cues of fibrillar myotubes [3] or platelets [4] related in the ALS clinical pathologies.

[2] Magazzù, A.; Marcuello, C. Investigation of Soft Matter Nanomechanics by Atomic Force Microscopy and Optical Tweezers: A comprehensive Review. Nanomaterials 2023, 13, 963. https://doi.org/10.3390/nano13060963.

[3] Varga, B.; Martin-Fernandez, M.; Hilaire, C.; Sanchez-Vicente, A.; Areias, J.; Salsac, C.; Cuisinier, F.J.G.; Raoul, C.; Scamps, F.; Gergely, C. Myotube elasticity of an amyotrophic lateral sclerosis mouse model. Sci. Rep. 2018, 8, 5917. https://doi.org/10.1038/s41598-018-024027-5.

[4] Strijkova, V.; Todinova, S.; Andreeva, T.; Langari, A.; Bogdanova, D.; Zlatareva, E.; Kalaydzhiev, N.; Milanov, I.; Taneva, S.G. Platelet’s Nanomechanics and Morphology in Neurodegenerative Pathologies. Biomedicines 2022, 10, 2239. https://doi.org/10.3390/biomedicines10092239.

7) REFERENCES. The references are in the proper format style of Biomedicines. No actions are requested from the authors.

The authors need to revise the manuscript in order to polish final details. Minor editing of English language is required.

Author Response

Comments responses for Reviewer 2

  1. Comment: TITLE. (OPTIONAL) The authors should consider add the full name “amyotrophic lateral sclerosis” instead of its abbreviated form.

Response: Thanks reviewer’s suggestion regarding the abbreviation issue in a title. We totally agree to use the full name of a disease instead of only using abbreviation. In response to your suggestion and another reviewer’s comments, the manuscript title has been changed to a new one: “Detection of Amyotrophic Lateral Sclerosis (ALS) Comorbidity Trajectories based on Principal Tree Model Analytics”.

  1. Comment: INTRODUCTION. “Amyotrophic Lateral Sclerosis (ALS) (…) is approximately three years” (lines 30-33). Here, it lacks a relevant reference related to the information content provided by the authors [1].

[1] Mead, R.J.; Shan, N.; Reiser, H.J.; Marshall, F.; Shaw, P.J. Amyotrophic lateral sclerosis: a neurodegenerative disorder poised for successful therapeutic translation. Nat. Rev. Drug Discov202322, 185-212. https://doi.org/10.1038/s41573-022-00612-2.
Response: Thank reviewer for pointing out the missing reference in the introduction section. We have added the suggested reference [1] to support the appropriate information about ALS.

  1. MATERIALS & METHODS. “This study anonymized data from 83 patients (…) over an 18-years span (1996-2013)”. Please, the authors should provide further statistic insights about the sex and age of the individuals who have participated in this study.

Response: Thank reviewer for emphasizing the importance of demographic details in the dataset. We have provided additional statistics for a better understanding of the dataset. The revised statement is as the followings:

… “This study utilized anonymized data from 83 patients (…) over an 18-year span (1996-2013) [22]. Of these patients, 51 (61.45%) were male and 32 (38.55%) were female. The age of their first ALS diagnosis ranged from 34 to 70 years, with an interquartile range (IQR) between 44 and 62 years.“ ….

  1. Comment: Figure 1 (line 109). The lettering related to this Figure should be enlarged. This will significantly aid potential readers to better visualized the information furnished by the authors. Same comment for Figure 2 (line 228) and Fig. 3 (line263).

Response: Thank reviewer for drawing attention to the legibility of the figures in our manuscript. We have enlarged the letters within Figures 1, 2, and 3 to enhance clarity and readability.

  1. Comment: Table 1 (line 193). Why did not the authors list the disease numbers by ICD-9 numerical order?

Response: Thank reviewer for the insightful comment. It is absolutely correct that ordering the diseases by ICD-9 numerical order would provide a more straightforward reference for readers. The original intention was to list them based on the Odds Ratio (OR) as a way of highlighting disease relevance. However, considering the table has shown the filtered diseases based on OR, this approach may not be the most intuitive. We have revised Table 1 and listed the associated diseases by their ICD-9 numerical order to enhance clarity and ease of reference for our readers. We appreciate your advice, which would improve the presentation of our data.

  1. Comment: DISCUSSION. The authors perfectly highlights the most relevant outcomes found in this work. Nevertheless, since this study is mainly focused on computational studies the authors need to add some experimental approaches which could be complementary with this current research. In this context, the use of single-molecule techniques [2] is relevant to unravel the nanomechanical fingeprint cues of fibrillar myotubes [3] or platelets [4] related in the ALS clinical pathologies.

[2] Magazzù, A.; Marcuello, C. Investigation of Soft Matter Nanomechanics by Atomic Force Microscopy and Optical Tweezers: A comprehensive Review. Nanomaterials 202313, 963. https://doi.org/10.3390/nano13060963.

[3] Varga, B.; Martin-Fernandez, M.; Hilaire, C.; Sanchez-Vicente, A.; Areias, J.; Salsac, C.; Cuisinier, F.J.G.; Raoul, C.; Scamps, F.; Gergely, C. Myotube elasticity of an amyotrophic lateral sclerosis mouse model. Sci. Rep20188, 5917. https://doi.org/10.1038/s41598-018-024027-5.

[4] Strijkova, V.; Todinova, S.; Andreeva, T.; Langari, A.; Bogdanova, D.; Zlatareva, E.; Kalaydzhiev, N.; Milanov, I.; Taneva, S.G. Platelet’s Nanomechanics and Morphology in Neurodegenerative Pathologies. Biomedicines 202210, 2239. https://doi.org/10.3390/biomedicines10092239.

Response: We really appreciate reviewer’s insightful comments. We agree that while computational models could offer unique insights, experimental data remain crucial for a comprehensive understanding and validation, especially in a complex field like neurodegenerative disorders.

To this end, we have incorporated into our discussion the significance of soft matter nanomechanics, specifically drawing from the groundbreaking work of Magazzù et al. [2] which emphasizes the utility of atomic force microscopy (AFM) and optical tweezers in probing single-molecule biomechanics. Furthermore, by integrating findings from Strijkova et al. [4], we discuss the potential of AFM in discerning mechanical and morphological imprints of platelets across diverse neurodegenerative conditions, underscoring the importance of such experimental methods in contextualizing and complementing computational results. The study by Varga et al. [3] further solidified this point by illustrating the potential insights garnered from examining the elasticity of ALS myotubes derived from specific mutations.

  1. Comment: REFERENCES. The references are in the proper format style of Biomedicines. No actions are requested from the authors.

Response: Thanks for reviewer’s comments on Reference section. We do appreciate for all your comprehensive reviews and helpful suggestions again.

Reviewer 3 Report

An excellent study that will assist with furthering the discussion of the development of more sophisticated and precise approaches to diagnose ALS.

I only have a few minor grammatical edits required.

Line 37: Full stop after reference.

Line 38: “…..physiology of ALS remains unknown [1].”

Line 39: “…remain elusive,….”

Line 42: “…he general population [6] .”

Line 63: “We introduce a novel…”

Author Response

An excellent study that will assist with furthering the discussion of the development of more sophisticated and precise approaches to diagnose ALS. Comments on the Quality of English Language, I only have a few minor grammatical edits required.

  1. Comment: Line 37: Full stop after reference.

Response: Thank reviewer for pointing out the error. We have added a full stop after the reference as suggested.

  1. Comment: Line 38: “…..physiology of ALS remains unknown [1].”

Response: Thank reviewer’s comment. We have revised it according to the suggestion.

  1. Comment: Line 39: “…remain elusive,….”

Response: Thanks for pointing it out. We have made the adjustment.

  1. Comment: Line 42: “…he general population [6] .”

Response: Thank reviewer’s comment. We have corrected the error. We appreciate your feedback.

  1. Comment: Line 63: “We introduce a novel…”

Response: Thank reviewer’s comment. We have corrected it. We appreciate all your feedback.